# A New Class of Benchmarks for Federated Multi-Objective Learning

## Abstract

Federated Learning allows effective machine learning in distribution without exposing the underlying training data. One emerging direction of research is the combination of Federated Learning with multi-objective methods, capturing the complexities of real-world problems by enabling training across multiple metrics of success even where such metrics conflict. The evaluation of novel methods requires suitable benchmarks. In Federated Learning, benchmarks are commonly transferred from centralized settings without modification. In this work, we show that this practice is not always sufficient: in one natural setting, where federated clients have heterogeneous preferences over multiple objectives, the most commonly used class of benchmarks can be solved easily even by baseline algorithms, in apparent contrast to the difficulty of the problem in the non-federated setting. Following this insight, we introduce a different, more challenging class of benchmarking problems, derived from the field of fair machine learning (fair ML). These benchmarks are adaptable, easy to implement, permit diverse model architectures and different (numbers of) objectives, include a range of different well-established datasets and do not require special adaptation of the federated algorithm. We run state-of-the-art algorithms on several instances of our proposed benchmarks, showing their versatility and applicability to a range of common Federated Multi-Objective Learning scenarios.

## 1 Introduction

Federated learning (FL) permits collaborative model training across distributed datasets that cannot be shared directly due to privacy constraints, confidentiality concerns or communication limitations. Instead of centralizing data, FL trains local models on each dataset and periodically exchanges parameters, which are then aggregated into a global model (McMahan et al., 2017). Since its inception, FL has rapidly expanded to a wide range of applications, including medical and financial settings, personal mobile devices, distributed tuning of large language models, and the Internet of Things (Guan et al., 2024; Antunes et al., 2022; Yang et al., 2021; Long et al., 2020; Ye et al., 2024; Nguyen et al., 2021). As these applications mature, more difficult problems emerge, driving research into advanced federated learning strategies (Kairouz et al., 2021).

One such direction is Federated Multi-Objective Learning (FMOL), which seeks to solve problems involving multiple, often conflicting, objectives in a federated setting. Multi-objective approaches capture the complexities of the real world more accurately, enabling algorithms to trade off competing criteria and identify balanced solutions, an aspect equally relevant in decentralized learning scenarios. Progress in FMOL critically depends on benchmarks that represent genuine conflicts between objectives. Existing benchmarks are primarily based on multi-task classification problems, where tasks can often be optimised jointly without inherent conflict, as we demonstrate in this work. As a result, they fail to represent the full spectrum of difficulty in multi-objective problems, limiting their utility for evaluating FMOL algorithms. This paper makes three key contributions:

**Benchmark analysis:** We show that widely used multi-task benchmarks fail to exhibit the expected trade-offs under federated settings, even when applying simple methods such as FedAvg.

**Fairness-based benchmarks:** To address this limitation, we introduce an alternative class of FMOL benchmarks built upon well-established fairness metrics from the fair ML literature. These benchmarks naturally encode conflicting objectives while remaining simple and flexible to construct and apply.

**Empirical validation:** Through experiments with both baseline and state-of-the-art FMOL algorithms, we demonstrate that fairness-based benchmarks reveal genuine multi-objective behavior, thereby providing a more meaningful testbed for algorithmic evaluation.

## 2 RELATED WORK

Federated Multi-objective Learning has only recently emerged as a dedicated direction of research, with relatively few works exploring general solution algorithms (Yang et al., 2023; Askin et al., 2025; Hartmann et al., 2025). To the best of our knowledge, the benchmarking of FMOL algorithms has not yet been explicitly addressed in the literature. In the absence of dedicated benchmarks, a common practice in the Federated Learning domain is to re-purpose existing ones from the centralized setting. In this vein, several works on FMOL (Yang et al., 2023; Askin et al., 2025) employ Multi-Task datasets, originally designed for benchmarking centralized multi-task learning (MTL) and multi-objective learning algorithms, to evaluate their methods.

One of the most commonly used datasets, both in federated and non-federated settings, is Multi-MNIST (Sener & Koltun, 2018). Multi-MNIST is constructed by combining two overlapping MNIST digits at an offset into a single image and concatenating the associated labels into an ordered sequence. The two tasks consist of classifying the left-most and right-most digit, respectively. Similar datasets constructed analogously include Multi-Fashion (Lin et al., 2019), combining two Fashion-MNIST images; Fashion-MNIST (Lin et al., 2019), combining samples from MNIST and Fashion-MNIST, respectively; and CIFAR-MNIST (Choe et al., 2020), combining CIFAR10 and MNIST images. In addition to such newly constructed datasets, pre-existing multi-label classification benchmarks such as the CelebA dataset (Liu et al., 2015) have also been used. In such works, straightforward learning approaches demonstrated an apparent trade-off between tasks.

A handful of works use other types of multi-objective problems for validation. Kinoshita et al. (2024) focus on solving unsupervised multi-objective optimisation problems such as clustering, which does not readily extend to general FL scenarios. Hartmann et al. (2025) use existing multi-objective reinforcement learning (MORL) benchmarks. While these represent problems that have definite, intuitively confirmable inherent trade-offs, sequential learning problems remain understudied in FL and thus should not be considered generally representative of the problem space.

Finally, some works also confront domain-specific or otherwise more narrowly defined problems that are multi-objective, without considering the general applicability. Fair federated learning is a line of research focused on ensuring fairness both between clients (Hu et al., 2022; Ju et al., 2024), and in the resulting model (Mehrabi et al., 2022). Though this abstraction is not remarked upon, the latter problem is multi-objective, as fairness and accuracy are known to conflict on biased datasets. None of these domain-specific problems present an immediately compelling alternative to the established MTL problems for general benchmarking purposes. However, we argue in the next section that the sole reliance on MTL problems is nevertheless suboptimal, as this class of benchmarks is likely not representative of the full difficulty of FMOL.

## 3 MULTI-TASK BENCHMARKS IN FEDERATION

In this section, we demonstrate the observation that served as the motivation for this work: that in certain natural settings, solving multi-task problems in federation appears to reduce or remove the conflict between individual tasks, simplifying the problem considerably from a multi-objective perspective. Conceptually, the difficulty of solving multi-objective problems arises from conflict between the individual objectives, where optimizing one objective reduces the utility of another. To accurately assess the performance of multi-objective algorithms, benchmarking problems should reflect this challenge. In some domains, the inherent conflict between objectives is immediately obvious. For example, a route planning algorithm for an autonomous vehicle may be expected to both minimize fuel usage and minimize the travel time to a given destination. Both objectives cannot generally be satisfied at the same time, as traveling faster consumes more fuel. In other domains, however, determining whether conflict between objectives is inherent or not is much more difficult. This is the case for the classification problems that are most often considered in FL. In multi-task benchmarks, non-federated experiments have shown a trade-off between the two objectives when assigning different fixed preferences to each, indicating that improvement in one objective harms

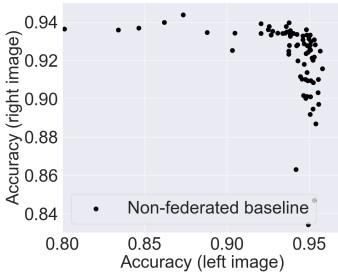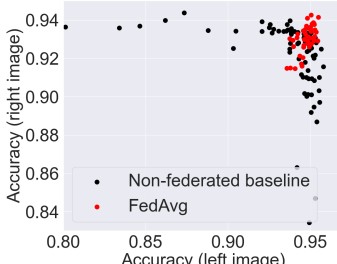

Figure 1: Results for non-federated (left) and with federated experiments (right) on Multi-MNIST with heterogeneous fixed preferences. Non-federated results show an apparent trade-off between the two objectives, but federated results do not. Federated results outperform non-federated ones, despite the forced collaboration between clients with different objective preferences.

the other. Yet it is not immediately clear that this is caused by an inherent conflict between the objectives. The model architecture used to solve such problems typically consists of a shared block, followed by individual model segments for each task. Given this architecture, and the independent nature of the two parallel classification tasks, there is no apparent reason why a sufficiently expressive network should not be able to separate the tasks and so satisfy both. Indeed, we speculate that the observed trade-off behavior may be a limitation of the learning algorithm caused by a lack of exploration of the parameter space, not a characteristic of the underlying learning problem. While the general absence of a conflict in MTL benchmarking datasets is difficult to prove, we give a motivating example that shows the apparent collapse of the Multi-MNIST benchmark in a particular federated use case: the preference-heterogeneous setting.

Preference heterogeneity has not yet received much attention in the literature, but is nonetheless a natural setting. It may occur in any use case where clients are self-interested: when training personalized user recommender systems, foundation models on proprietary data across multiple enterprises, or on-line route planners on autonomous vehicles. In this setting, each participating client has its own preferences regarding the importance of individual objectives. On problems with conflicting objectives, we would expect this to cause complications in the federated aggregation step: diverging local training trajectories may be difficult to reconcile. However, we observe a different result: In our experiments, the non-federated baseline does reproduce a set of trade-off solutions, or *Pareto front*, but the FedAvg algorithm yields better results with far less apparent trade-off. FedAvg is not known for handling heterogeneity well; yet this type of heterogeneity on this problem appears to improve the output significantly, removing conflict between objectives. We speculate that federated preference heterogeneity has the same effect as intentionally varying preference weights during the learning process, an approach that is employed intentionally in the design of more sophisticated algorithms solving (non-federated) MTL. A notable example is (Sener & Koltun, 2018), where such an algorithm generates a (single, arbitrary) solution that dominates a Pareto front generated with fixed weights – similar to what we observe here. From these considerations, we conclude that standard MTL benchmarks may not be a challenging benchmark for FMOL algorithms. While the general absence of a conflict in this and other MTL benchmarking datasets is difficult to prove, we consider this argument compelling enough to propose the use of additional classes of benchmarks in combination with MTL datasets. In the remainder of this paper, we propose a different class of multi-objective (multi-criteria) problems as benchmarks and, using these problems, demonstrate that MOO remains a challenge in federation.

## 4 DESIGNING ALTERNATIVE BENCHMARKS – GROUP FAIRNESS

Based on the observations outlined in the previous section, we argue that more challenging benchmarking problems are needed to comprehensively evaluate the performance of FMOL algorithms. To address this need, we introduce a new class of benchmarks constructed by adapting established problems from the field of fair machine learning into generally applicable problem formulations. This section first outlines key concepts in fair ML before detailing our proposed benchmarks.

## 4.1 BACKGROUND: FAIRNESS IN MACHINE LEARNING

Many real-world datasets, particularly those involving demographic data, are known to contain imbalances that reflect underlying cultural or societal biases. Examples include racial disparities in criminal sentencing decisions, gender-based differences in income determination, and age-related biases in health records. Training prediction models on such datasets risks propagating these biases, leading to discriminatory behavior in automated decision-making systems. A well-known example is the COMPAS dataset used for recidivism prediction, which has been shown to systematically discriminate against Black defendants (Angwin et al., 2016).

Bias mitigation has been extensively studied, with existing approaches typically categorized into three families: debiasing the underlying dataset (pre-processing), preventing the learning of biases during training (in-processing), and modifying the output of the trained model to enhance fairness (post-processing) (Mehrabi et al., 2021). Of particular interest here are in-processing methods that can formulate the learning process as a multi-objective problem, introducing fairness as additional objectives. This formulation serves as the foundation for our proposed benchmarks.

Fairness in machine learning has been formalized through various metrics, often formulated with respect to a binary *sensitive attribute*, quantifying disparities in predicted outcomes between sub-populations (Mehrabi et al., 2021). A classifier is considered perfectly fair if outcomes are statistically indistinguishable across these groups. Such group fairness metrics include demographic parity (Dwork et al., 2012; Kusner et al., 2017), equality of opportunity (Hardt et al., 2016), and equalized odds (Hardt et al., 2016).

**Demographic parity** (DP) requires the overall probability of a positive classification outcome, such as loan approval, to be equal between the in-group and the out-group. Let $X$ be the set of input data, $Y$ the set of labels and $S$ the labels of sensitive attributes. In formal terms, a classifier satisfies demographic parity if, across all predictions,

$$P(\hat{y} = 1 | s = 0) = P(\hat{y} = 1 | s = 1), \tag{1}$$

where $\hat{y}$ is the binary predicted outcome, and $s \in S$ is the sensitive attribute.

**Equality of opportunity** (EO) demands equal probabilities of *true* positive outcomes between groups, i.e.

$$P(\hat{y} = 1 | s = 0, y = 1) = P(\hat{y} = 1 | s = 1, y = 1), \tag{2}$$

where $y \in Y$ is the ground-truth label of a given sample.

**Equalized odds** (EOD) requires equal probabilities of true positive as well as false positive outcomes across groups:

$$P(\hat{y} = 1 | s = 0, y = 1) = P(\hat{y} = 1 | s = 1, y = 1)$$
$$\wedge P(\hat{y} = 1 | s = 0, y = 0) = P(\hat{y} = 1 | s = 1, y = 0). \tag{3}$$

For practical use as a fairness score on classification data, these definitions can be reformulated as the stochastic difference between the left- and right-hand side of the equation, e.g. for DP, we formalize the Difference of Demographic Parity (DDP) as follows:

$$DDP(X, Y, S, f)$$
$$= \frac{1}{n_{s=0}} \sum_{\substack{x \in X \\ s=0}} [f(x) > 0.5] - \frac{1}{n_{s=1}} \sum_{\substack{x \in X \\ s=1}} [f(x) > 0.5],$$

where $f : X \to [0, 1]$ is the predictor, $[f(x) \to 0, 1]$ is the binary classification decision and $n_{s=s'}$ is the number of samples with sensitive attribute value $s' \in \{0, 1\}$. The existence of *fairness impossibility* is a is a well-known result in fair ML, stating that certain fairness concepts, including demographic parity, cannot be jointly optimized with error-based metrics on biased datasets. A full overview of existing fairness metrics, together with a comprehensive discussion of theoretical and empirical incompatibility results, can be found in (Pessach & Shmueli, 2022).

## 4.2 FORMULATION OF BENCHMARKING PROBLEM

Using these established metrics, we construct straightforward additional benchmarks. We focus on ease of implementation and evaluation, discarding more subtle design choices in favor of simplicity.

### 4.2.1 OBJECTIVES

The stochastic formulation of the fairness metrics described above is not differentiable, and as such not well-suited for direct use as a loss function in stochastic gradient descent. Various relaxation approaches have been proposed, e.g. (Lohaus et al., 2020; Celis et al., 2019), often in combination with

| Dataset | Description | Sens. attrs. |
|---|---|---|
| Adult Income (Becker & Kohavi, 1996) | Demographic data, predicting binary income class | Gender, Race |
| Law School (Wightman, 1998) | Demographic and academic data, predicting bar passage | Gender, Race |
| Credit Default (Yeh & Lien, 2009) | Demographic and financial data, predicting credit card default | Gender, Age |
| Compas (Angwin et al., 2016) | Demographic and criminal history data, predicting recidivism | Race |
| CelebA (Liu et al., 2015; Denton et al., 2019) | Multi-label image classification of faces | Gender |
| Heritage Health (Goldbloom & Hamner, 2011) | Demographic and health data, predicting hospitalization | Age |

Table 1: Selection of common benchmarking datasets in the fair ML domain, all usable with our proposed formulation as drop-in benchmarks for FMOL algorithms.

specific solution algorithms such as constraint-based optimization. Two other works, focused specifically on fair federated learning, side-step the problem by using fairness metrics only as a secondary scoring mechanism(Mehrabi et al., 2022) or optimization constraint (Cui et al., 2021). None of these formulations generalizes readily into an abstract multi-objective problem that admits different solution approaches, such as stochastic multi-gradient descent. Instead, we propose to use a recently introduced relaxation method that yields a differentiable approximation of the metric (Padh et al., 2021), directly usable as a loss function. The hyperbolic tangent relaxation is straightforwardly applicable to most standard group fairness metrics, including those presented here, is model-agnostic, and can be be used for multiple metrics simultaneously. This gives us scalability, allowing the design of many-objective problems, and flexibility in choosing the local learning approach. Under the tanh relaxation, the prediction $f(x) : X \to [0, 1]$ of the classifier is relaxed to

$$\hat{f}(x) = tanh(c \cdot max(0, 2f(x) - 1))/2 + 0.5, \tag{4}$$

where $c \in \mathbb{R}$ regulates the trade-off between the precision of the approximation and the behavior of the gradient[1]. The relaxed prediction is then used in place of the binary result in computing the chosen fairness metric, e.g. for the DDP metric:

$$\widehat{DDP}(X, Y, S, \hat{f}) = \frac{1}{n_{s=0}} \sum_{\substack{x \in X \\ s=0}} \hat{f}(x) - \frac{1}{n_{s=1}} \sum_{\substack{x \in X \\ s=1}} \hat{f}(x) \tag{5}$$

We use the relaxed gap metric for one or more fairness metrics as individual objectives in a multi-objective problem, combined with the accuracy objective. For the accuracy objective, an appropriate loss function for the learning problem is used. Note that many fairness benchmarking datasets have more than one sensitive attribute, e.g. gender and race, with attributes mutually independent. This provides a straightforward avenue for the construction of problems with more than two objectives. Similarly, multiple different fairness metrics can be applied simultaneously as separate objectives, with various group fairness metrics known to mutually conflict (Castelnovo et al., 2022).

Another benefit of this fairness formulation is flexibility w.r.t. the local learning strategy. Some FMOL algorithms specify a local learning algorithm, e.g. multi-gradient descent (Yang et al., 2023; Askin et al., 2025); others, e.g. (Hartmann et al., 2025), do not. This formulation of the fairness problem is equally accessible to all these algorithms, and even allows the testing of algorithms not specific to the setting, e.g. the FedAvg baseline algorithm (McMahan et al., 2017).

### 4.2.2 DATASETS

A great number of different benchmarking datasets in varying size and shape exist in the domain of fair ML. All such datasets that contain sensitive attributes can be used with the loss functions defined here. The vast majority is based on real-world data collected for other purposes, with underlying biases identified by later research. This is an advantage for benchmarking, as datasets represent tangible and realistic use cases, many with obvious relevance to the federated setting. Though this

---

[1]This definition differs slightly from that given in the reference paper, which was based on predictions in the range of [-1, 1].

real-world origin can also present challenges, such as flawed or incomplete data, many frequently used datasets are available in cleaned form. We list a selection of commonly used datasets in Table 1. For a more comprehensive overview of datasets we refer to the appendix of the survey by Pessach & Shmueli (2022); a second survey (Le Quy et al., 2022) contains a particularly detailed description of several datasets.

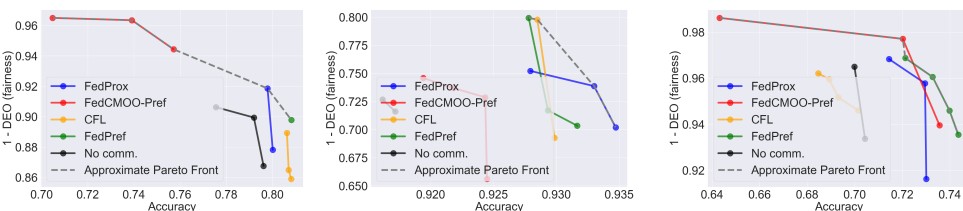

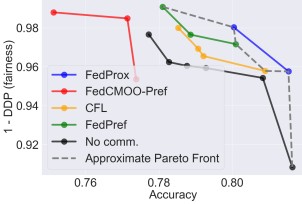 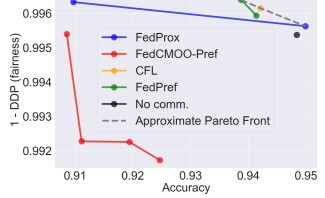 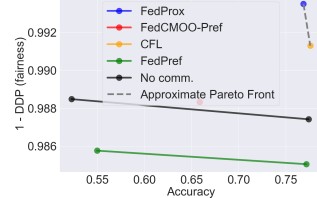

(a) Adult dataset with binarized race as the sensitive attribute and DEO fairness.

(b) Law School dataset with binarized race as the sensitive attribute and DEO metric.

(c) Default dataset with gender as the sensitive attribute and DEO metric.

(d) Adult dataset with binarized race as the sensitive attribute and DDP metric.

(e) Law School dataset with binarized race as the sensitive attribute and DDP metric.

(f) Default dataset with gender as the sensitive attribute and DDP metric.

Figure 2: Results of different algorithms on a selection of benchmark problems for accuracy and equality of opportunity (top row) and accuracy and demographic parity (bottom row). All clients were assigned the same preferences during a run, with 10 runs performed on preferences from $(0., 1.0)$ to $(0.9, 0.1)$, modified by steps of $(+0.1, -0.1)$. Each point represents the mean client output for a single run, with the Pareto fronts across all runs reported for each algorithm. All reported fairness metrics are inverted for ease of visualization, i.e. 1 corresponds to perfect fairness.

## 5 EXPERIMENTS

We demonstrate the validity and usability of the proposed class of fairness benchmarks by constructing ten bi-objective example problems, combining three different fairness datasets with a total of five sensitive attributes with two different fairness metrics. We select three common and readily available fairness benchmarking datasets:

**UCI Adult**: Data extracted from the 1994 US Census database. Using demographic information to predict whether a person's income exceeds $50,000$ per year. The sensitive attributes used in our experiments are gender and race (binarized into *white* and *not white*).

**Law School**: Data on US law students between 1991-1997. Using demographic information and earlier test scores to predict whether a candidate passes the bar. Sensitive attributes used are binarized race and gender.

**Credit Card Default**: Predicting from personal data and credit card history whether a bank customer will default on the next credit card payment. The sensitive attribute is the customer's gender. We use accuracy as one objective metric, and either the difference of demographic parity (DDP) or difference of equality of opportunity (DEO) as fairness metric to construct two distinct bi-objective problems on each dataset-attribute combination, defining the fairness loss as described in the previous section and the accuracy loss as binary cross-entropy with logits. We run four representative FL algorithms on these benchmarks: FedProx, CFL, FedCMOO, and FedPref. FedProx is a standard baseline algorithm performing centralized aggregation, with some tolerance for client heterogeneity (Li et al., 2020). CFL is a clustering-based algorithm that generates personalized client models (Sattler et al., 2021). Though originally designed for settings with incompatible client data, the adaptive clustering strategy may be applicable for multi-objective heterogeneity as well. Fed-

CMOO is a recent algorithm designed specifically for federated multi-objective learning (Askin et al., 2025). We implement the FedCMOO-Pref variant, equipped to handle homogeneous objective preferences. In contrast, FedPref, another recent FMOL algorithm, is intended specifically for federating preference-heterogeneous clients (Hartmann et al., 2025). Finally, we also run the standard non-federated baseline. All algorithms are tuned via grid search, with details reported in the appendix. We test two natural and distinct scenarios for federating multi-objective problems. First, we solve a multi-objective problem in collaboration between clients that all share the same objective preferences. Then, we run the setting where all clients have individual, heterogeneous objective preferences. The experiments reported here are run on systems of 10 clients; the appendix includes further experiments on up to 50 clients.

## 5.1 HOMOGENEOUS PREFERENCES

This setting corresponds to the multi-objective equivalent of the most common focus in FL, where clients collaborate to train a single global model that generalizes over all data available in distribution. To explore the multi-objective performance of algorithms on this benchmark, we generate a set of 10 equally-spaced preference weights. We run each benchmarking problem 10 times, once for each preference weight, and report the mean client results for each run. Following standard practice from multi-objective optimization, we compute the Pareto front of solutions per algorithm, and report the hypervolume metric for each with respect to the coordinate origin. We also show the minimum and maximum values found for each objective and algorithm in the appendix, illustrating the spread of results.

Figure 2 shows the Pareto fronts obtained for six of the benchmarking problems – three optimizing the DEO fairness metric, and three the DDP metric, with the corresponding hypervolume values reported in Table 2. A trade-off between the accuracy and fairness objective is readily apparent in all six plots, with different performances by the tested algorithms on different datasets. Nevertheless, some general observations can be noted: in almost all cases, all federated algorithms outperform the non-federated baseline. (An exception is the DDP metric on the Law School dataset, where the single solution reported for the baseline is most likely an outlier that did not converge. Statistical noise is a challenge for these datasets that is discussed in more detail at the end of this section.) The FedProx algorithm performs relatively well in this setting, indicating that, at least in the absence of objective heterogeneity, the basic algorithm is capable of finding some appropriate trade-off solutions. The other federated algorithms all show mixed performances: in Fig. 2a and Fig. 2c, Fed-CMOO notably explores sections of the Pareto front discovered by no other algorithm. A similar tendency, though less successful, appears in the Adult dataset with the DDP metric (Fig. 2d). The increased exploration range may be explained by the design of FedCMOO, which adaptively adjusts the initial objective preferences during the training process. The FedPref algorithm discovers at least one Pareto-dominant solution in five cases, but appears occasionally very limited in its exploration of the Pareto front (see e.g. Fig. 2a and 2e). It is possible that the clustering mechanism of the algorithm is counterproductive in this homogeneous setting, where no obvious groupings of clients exist. A similar issue may be limiting the performance of the CFL algorithm by separating clients where no inherent incompatibility exists. Finally, it should be noted that these results may not represent the true potential of each algorithm. In a thorough multi-objective evaluation of the algorithms, a heuristic would normally be employed to search the space of preference weights to generate a balanced Pareto front. In this work, intended mainly to demonstrate the usability of the proposed class of benchmarks, the exploration was instead restricted to a predefined set of preference weights.

## 5.2 HETEROGENEOUS PREFERENCES

This second setting represents a use case with high heterogeneity, where each client in the federation has individual preferences. Such a setting is commonly connected to Personalized Federated Learning (PFL) approaches, where the focus of the algorithm is shifted from generalized global to individual client performance. Instead of generating a single global model, the traditional aggregation approach is modified to yield a separate personalized model for each client, fitted to that client's unique characteristics. We run the same selection of algorithms as in the previous experiments. As some of these algorithms are not designed to generate personalized client models, we have included the option of a single fine-tuning step at the end of the training phase in the hyperparameter tuning. 10 preference sets were generated uniformly at random and submitted to each algorithm. We visualize the results in Fig.3 and report the corresponding hypervolumes in Table 3, with additional

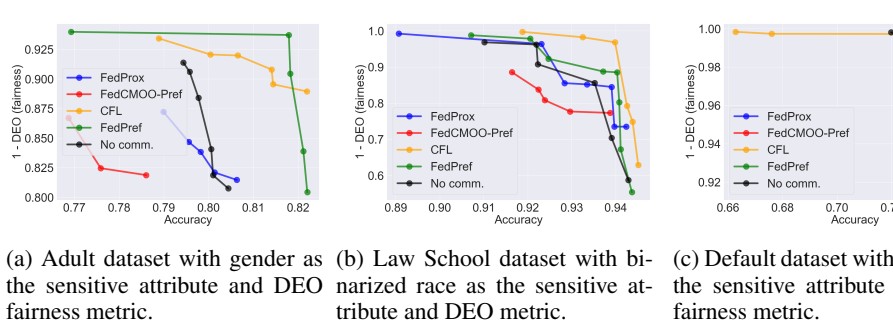

(a) Adult dataset with gender as the sensitive attribute and DEO fairness metric.

(b) Law School dataset with binarized race as the sensitive attribute and DEO metric.

(c) Default dataset with gender as the sensitive attribute and DEO fairness metric.

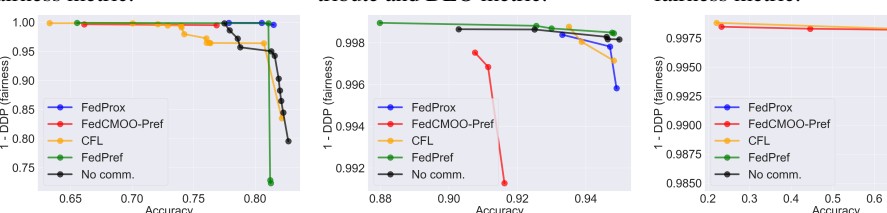

(d) Adult dataset with gender as the sensitive attribute and DDP fairness metric.

(e) Law School dataset with gender as the sensitive attribute and DDP metric.

(f) Default dataset with gender as the sensitive attribute and DDP metric.

Figure 3: Results of different algorithms on a selection of benchmark problems for accuracy and equality of opportunity (top row) and accuracy and demographic parity (bottom row). Clients were assigned heterogeneous preferences during each run, generated uniformly at random but the same across algorithms. Each point represents the output of a single client, with the Pareto fronts across all runs reported for each algorithm. All reported fairness metrics are inverted for ease of visualization, such that 1 corresponds to perfect fairness.

results, including for a greater number of clients, reported in the appendix. Unlike in the homogeneous setting, we do not average the results for each run, but instead consider the individual client solutions. Due to space limitations, the min-max analysis is once again included in the appendix.

We observe that the FedProx algorithm performs notably worse in the heterogeneous setting, particularly in experiments with the DEO metric – see e.g. its performance on the Adult dataset (Fig. 3a), where it is outperformed even by the non-federated baseline. This is consistent with our expectation that algorithms with centralized aggregation would struggle in preference-heterogeneous settings with conflicting objectives. The FedCMOO algorithm, too, is not designed for this setting, and perhaps has difficulties in reconciling incompatible clients. FedPref and CFL, the personalized algorithms, both do better in this setting. For both the homogeneous and the heterogeneous setting, we also observe that the behavior observed on the different datasets is quite consistent as the number of clients increases, as seen in the additional results reported in the appendix. Finally, we note that the DDP metric generally appears more difficult to solve than the DEO metric, with fewer points discovered on the Pareto front, and a tendency for those points to be extreme. The trade-off between DDP and accuracy may be more difficult to regulate, or learning trajectories diverge earlier,

| Data - | Accuracy - DEO | | | | | Accuracy - DDP | | | | |
|---|---|---|---|---|---|---|---|---|---|---|
| Sens. attr. | FProx | CFL | FCMOO | FPref | no comm | FProx | CFL | FCMOO | FPref | no comm |
| Adult - G | **0.701** | 0.687 | 0.676 | 0.696 | 0.678 | 0.763 | 0.788 | 0.754 | 0.761 | **0.805** |
| Adult - R | **0.735** | 0.719 | 0.730 | 0.726 | 0.721 | **0.799** | 0.792 | 0.764 | 0.793 | 0.796 |
| Law - G | 0.882 | 0.821 | **0.919** | 0.836 | 0.796 | **0.946** | 0.938 | 0.920 | 0.938 | 0.944 |
| Law - R | 0.703 | 0.742 | 0.689 | **0.744** | 0.667 | 0.911 | 0.944 | **0.945** | 0.928 | 0.940 |
| Default - G | 0.707 | 0.675 | **0.724** | 0.720 | 0.680 | 0.764 | **0.769** | 0.651 | 0.760 | 0.765 |

Table 2: Hypervolumes of global performance results for accuracy and DEO (left) and accuracy and DDP (right) on homogeneous preferences. Higher is better (Fairness metrics are inverted, as in the results figures). Only results from the algorithm-specific Pareto front are reported (see also Fig. 2)

allowing less time for collaboration between clients. This hypothesis may explain why even the more successful FL algorithms struggle to outperform the non-federated baseline in Fig. 3, and why there is relatively little diversity in the Pareto fronts discovered in the homogeneous setting in Fig.2. Finally, a comparison of the Pareto fronts with the corresponding hypervolume and min-max values reveals an interesting insight: while the potential values of the objective metrics cover the same interval, in practice the trade-off between the objectives plays out in different magnitudes on the two axes. Hence e.g. the highest overall hypervolume on the Adult dataset with gender as the sensitive attribute in Table 3 is achieved by the no-communication baseline, even though it does not appear obviously superior in the illustration in Fig. 3d. This imbalanced magnitude of metrics presents a challenge that is often encountered in the real world. As such, there is use in evaluating the ability of algorithms to cope with such problems, where the magnitude of objective gradients may differ.

| Data -      | Accuracy - DEO | | | | | Accuracy - DDP | | | | |
| Sens. attr. | FProx | CFL | FCMOO | FPref | no comm | FProx | CFL | FCMOO | FPref | no comm |
|---|---|---|---|---|---|---|---|---|---|---|
| Adult - G   | 0.703 | 0.767 | 0.681 | **0.772** | 0.734 | 0.814 | 0.816 | 0.766 | 0.811 | **0.822** |
| Adult - R   | 0.715 | 0.794 | 0.732 | 0.796 | **0.802** | 0.822 | 0.818 | 0.773 | **0.830** | 0.823 |
| Law - G     | 0.942 | **0.946** | 0.896 | 0.939 | 0.940 | 0.947 | 0.947 | 0.914 | 0.947 | **0.948** |
| Law - R     | 0.931 | **0.941** | 0.829 | 0.929 | 0.909 | 0.949 | 0.949 | 0.949 | 0.949 | **0.949** |
| Default - G | 0.722 | **0.745** | 0.745 | 0.742 | 0.736 | 0.750 | 0.780 | 0.722 | 0.786 | **0.793** |

Table 3: Hypervolumes of global performance results for accuracy and DEO (left) and accuracy and DDP (right) on heterogeneous preferences. Higher is better (Fairness metrics are inverted, as in the results figures). Only results from the algorithm-specific Pareto front are reported (see also Fig. 3).

### 5.3 PRACTICAL CONSIDERATIONS

Our implementation of these experiments is available on git[2]. Deploying this class of benchmarks in other existing implementations of federated algorithms requires minimal modifications in principle. Many fairness datasets are widely available and, as with those used here, can often be accessed directly through common ML libraries such as PyTorch. The local learning process does not generally need adjustment beyond the addition of the fairness loss functions and evaluation metrics, which are lightweight and easily portable. Tuning local learning parameters to explore diverse trade-offs can be challenging; we provide notes on parameter selection and implementation details in the appendix. Fairness datasets are often highly imbalanced with respect to sensitive and classification labels. In a FL context, this introduces two issues. First, non-converged models may return deceptively high accuracy and fairness values – in most fairness metrics, blanket predictions of a fixed value are by definition perfectly fair. Such results should be excluded in analyses and parameter searches – we describe a simple filtering strategy in the appendix. Secondly, some variability in client-level results, particularly under heterogeneous preferences, likely comes from naive random partitioning of data. Future benchmarks would benefit from splitting strategies that preserve the distribution of all pairs of labels and sensitive attributes across clients.

## 6 CONCLUSION AND FUTURE WORK

In this work, we have introduced a new class of benchmarks for evaluating Federated Multi-objective Learning algorithms, addressing the limitations of existing multi-task benchmarks that often lack genuine objective conflicts in federated settings. Conflicts between utility and fairness, as well as between different fairness metrics, are well-established in the domain of Fair ML. Our experiments confirm that our proposed fairness-based benchmarks are versatile, simple to implement, and capable of exposing meaningful trade-offs between objectives across various FL scenarios.

Rather than advocating for a complete replacement of current benchmarks, we argue for their diversification to better reflect the challenges of FMOL. Future work should investigate improved data partitioning strategies to reduce noise in fairness datasets and better control client heterogeneity.Finally, our findings suggest that heterogeneous client preferences may, paradoxically, facilitate

---

[2]Link removed for anonymity.

optimization by improving parameter space exploration in multi-task FL. Understanding and exploiting this effect could open new directions for distributed multi-task and multi-objective learning.

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

## A  APPENDIX

### A.1  MULTI-MNIST EXPERIMENTS

The motivational experiment presented in Section 3 contrasts the results generated by FedAvg and the non-federated baseline when run with the same hyperparameters (where applicable), listed in Table 4. These experiments use the same neural network architecture as the original paper proposing Multi-MNIST benchmarks Sener & Koltun (2018), based on a modified LeNet. As in the original paper, the top two layers are split off from the main network, with a separate top segment used for each task.

Both variants were run on a minimal federated system of 2 clients, where the two clients are assigned a contrasting preference distribution. Ten such configuration were generated, with preferences set to $[(0, 1), (1, 0)]$, $[(0.1, 0.9), (0.9, 0.1)]$, et cetera. Each preference configuration was run five times.

| Parameter | Value | Comment |
|---|---|---|
| Learning rate | $10^{-1}$ | |
| Total iterations | 100 | |
| Batch size | 256 | |
| Num. local iterations | 1 | |
| Random seed | $0, 1, 2, 3, 4$ | |

Table 4: Hyperparameters used for the motivational experiment presented in Section 3

| Algorithm | Parameter | Tested values | Comment |
|---|---|---|---|
| no comm | Learning rate | $5 \cdot 10^{-4}, 10^{-3}, 10^{-2}$ | No federated parameters. |
| FedProx | Learning rate | $5 \cdot 10^{-4}, 10^{-3}, 10^{-2}$ | |
| | Num. local iterations | $10, 25, 50$ | |
| | Proximal term $\mu$ | $0, 0.01, 0.1$ | $\mu = 0$ recovers standard FedAvg |
| | Finetuning rounds | $0, 1$ | |
| CFL | Learning rate | $5 \cdot 10^{-4}, 10^{-3}, 10^{-2}$ | |
| | Num. local iterations | $10, 25, 50$ | |
| | Clustering threshold | $1, 2.5, 5, 7.5$ | |
| | Patience | $1, 2$ | Rounds below threshold before clustering triggered[a] |
| | Finetuning rounds | $0, 1$ | |
| FedCMOO | Learning rate | $5 \cdot 10^{-4}, 10^{-3}, 10^{-2}$ | |
| | Global learning rate | $1.0, 1.5, 2.0, 2.5$ | |
| | Num. local iterations | $10, 25, 50$ | |
| | Finetuning rounds | $0, 1$ | |
| FedPref | Learning rate | $5 \cdot 10^{-4}, 10^{-3}, 10^{-2}$ | |
| | Num. local iterations | $10, 25, 50$ | |
| | Clustering threshold | $1, 2.5, 5, 7.5$ | Relative change |
| | Patience | $1, 2$ | Rounds below threshold before clustering triggered[a] |
| | Finetuning rounds | $0, 1$ | |

[a] Introduced by us to handle slow initial gradient ramp-up and noise introduced by client heterogeneity.

Table 5: Complete list of parameter configurations tested during hyperparameter tuning of algorithms.

## A.2 EXPERIMENTAL CONFIGURATION

Our model architecture for all experiments consists of a simple neural network with two hidden layers of size $64$ and $32$, respectively, using ReLU activation functions. For the output layer we use a Sigmoid activation function.

## A.3 PARAMETER TUNING

We tune all algorithms by grid search, running each configuration for three runs. The tested parameter values are listed in Table 5, with values that were ultimately selected shown in Table 6 and Table 7. The same three heterogeneous preference assignments were tested for each configuration across algorithms, with each set of preferences drawn uniformly at random from the weight simplex. In evaluating the results of the parameter search, we observe a Pareto front, with different configurations producing different trade-off solutions.

## A.4 ADDITIONAL RESULTS

This section contains the experimental results that were discussed in the main paper, but could not be reported in detail. The remaining plots of Pareto fronts found by different algorithms on 10 clients

| Algorithm | Parameter | AD - G | AD - R | LS - G | LS - R | DFT |
|---|---|---|---|---|---|---|
| no comm | Learning rate | $5 \cdot 10^{-4}$ | $10^{-3}$ | $10^{-3}$ | $10^{-3}$ | $10^{-3}$ |
| FedProx | Learning rate | $10^{-3}$ | $10^{-3}\ 10^{-2}$ | $10^{-2}$ | $10^{-3}$ | |
| | Num. local iterations | 50 | 50 | 25 | 25 | 50 |
| | Proximal term $\mu$ | 0 | 0.01 | 0 | 0.01 | 0 |
| | Finetuning rounds | 0 | 0 | 1 | 1 | 0 |
| CFL | Learning rate | $10^{-2}$ | $10^{-2}$ | $10^{-2}$ | $10^{-2}$ | $10^{-2}$ |
| | Num. local iterations | 25 | 50 | 50 | 50 | 25 |
| | Clustering threshold | 7.5 | 5 | 7.5 | 5 | 7.5 |
| | Patience | 1 | 1 | 2 | 2 | 2 |
| | Finetuning rounds | 1 | 1 | 0 | 0 | 1 |
| FedCMOO | Learning rate | $10^{-2}$ | $10^{-2}$ | $10^{-3}$ | $10^{-2}$ | $10^{-2}$ |
| | Global learning rate | 2.0 | 2.5 | 2.5 | 1.5 | 2.5 |
| | Num. local iterations | 10 | 50 | 50 | 10 | 25 |
| | Finetuning rounds | 0 | 0 | 0 | 1 | 0 |
| FedPref | Learning rate | $10^{-2}$ | $10^{-2}$ | $10^{-2}$ | $10^{-2}$ | $10^{-3}$ |
| | Num. local iterations | 50 | 25 | 25 | 50 | 50 |
| | Clustering threshold | 7.5 | 2.5 | 1.0 | 1.0 | 1.0 |
| | Patience | 1 | 1 | 2 | 1 | 2 |
| | Finetuning rounds | 0 | 0 | 1 | 1 | 0 |

Table 6: Parameter configurations selected for each algorithm and problem with the DEO fairness metric. Left to right: Adult dataset with gender as sensitive attribute, adult - race, Law School - gender, Law school - race, Default -gender.

| Algorithm | Parameter | AD - G | AD - R | LS - G | LS - R | DFT |
|---|---|---|---|---|---|---|
| no comm | Learning rate | $5 \cdot 10^{-4}$ | $5 \cdot 10^{-4}$ | $10^{-3}$ | $10^{-3}$ | $5 \cdot 10^{-4}$ |
| FedProx | Learning rate | $10^{-2}$ | $10^{-2}\ 10^{-3}$ | $10^{-3}$ | $10^{-2}$ | |
| | Num. local iterations | 25 | 25 | 10 | 10 | 10 |
| | Proximal term $\mu$ | 0 | 0.1 | 0 | 0.1 | 0.01 |
| | Finetuning rounds | 1 | 1 | 0 | 0 | 1 |
| CFL | Learning rate | $10^{-3}$ | $10^{-3}$ | $10^{-2}$ | $10^{-2}$ | $10^{-3}$ |
| | Num. local iterations | 25 | 50 | 25 | 50 | 50 |
| | Clustering threshold | 1 | 1 | 5 | 7.5 | 2.5 |
| | Patience | 2 | 1 | 1 | 2 | 2 |
| | Finetuning rounds | 1 | 0 | 0 | 0 | 1 |
| FedCMOO | Learning rate | $10^{-2}$ | $10^{-2}$ | $10^{-2}$ | $5 \cdot 10^{-4}$ | $5 \cdot 10^{-4}$ |
| | Global learning rate | 2.0 | 1.0 | 1.5 | 1.0 | 2.5 |
| | Num. local iterations | 50 | 25 | 50 | 10 | 25 |
| | Finetuning rounds | 0 | 0 | 0 | 0 | 0 |
| FedPref | Learning rate | $10^{-2}$ | $10^{-2}$ | $10^{-2}$ | $10^{-2}$ | $10^{-3}$ |
| | Num. local iterations | 50 | 10 | 50 | 50 | 10 |
| | Clustering threshold | 2.5 | 1.0 | 5.0 | 5.0 | 7.5 |
| | Patience | 1 | 2 | 1 | 1 | 1 |
| | Finetuning rounds | 1 | 1 | 1 | 1 | 0 |

Table 7: Parameter configurations selected for each algorithm and problem with the DDP fairness metric. Left to right: Adult dataset with gender as sensitive attribute, Adult - race, Law School - gender, Law school - race, Default -gender.

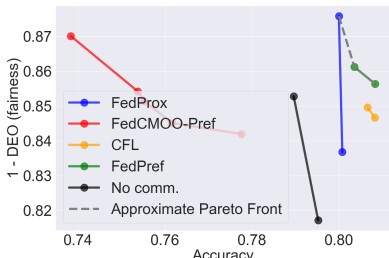

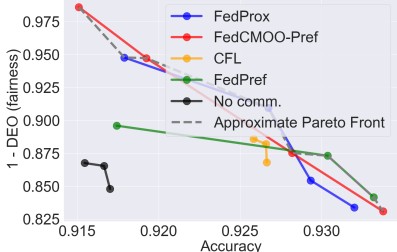

(a) Adult dataset with gender as the sensitive attribute and DEO fairness metric.

(b) Law School dataset with gender as the sensitive attribute and DEO fairness metric.

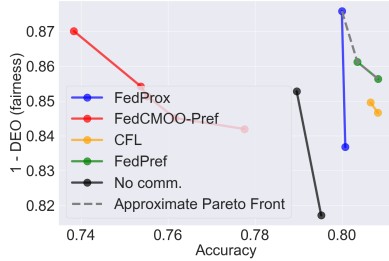

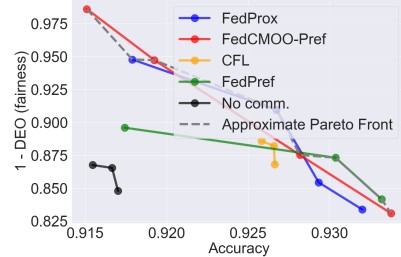

(c) Adult dataset with gender as the sensitive attribute and DDP fairness metric.

(d) Law School dataset with gender as the sensitive attribute and DDP fairness metric.

Figure 4: Results of different algorithms on 10 clients on a selection of benchmark problems for accuracy and equality of opportunity (top row) and accuracy and demographic parity (bottom row). All clients were assigned the same preferences during a run, with 10 runs performed on preferences from $(0., 1.0)$ to $(0.9, 0.1)$, modified by steps of $(+0.1, -0.1)$. Each point represents the mean client output for a single run, with the Pareto fronts across all runs reported for each algorithm. All reported fairness metrics are inverted for ease of visualization, such that 1 corresponds to perfect fairness.

are shown in Fig. 4 and Fig. 5 for experiments with homogeneous and heterogeneous preferences, respectively. The corresponding minimum and maximum values for each metric and experiment can be found in Tables 8, 9, 10, and 11.

In addition to the experiments presented in the main section, we have also carried out additional experiments scaled to federated systems of 50 clients. These results are visualised in Figs. 6 and 7 and Figs. 8 and 9 for homogeneous and heterogeneous preferences, respectively. The corresponding numerical hypervolume values may be found in Tables 12 and 15, with the minimum and maximum values reported in Tables 13, 14, 16, and 17.

## A.5 PRACTICAL REMARKS

In the main paper, we have noted the presence of statistical noise in client results. With multi-objective analysis in particular, outliers could distort the reported performance of algorithms, e.g. if identified as points on the Pareto front. Therefore, it may be useful to account for this noise in multi-objective analysis, e.g. by relaxing the strict Pareto front to one of rank $k$ as defined in (Deb et al., 2002)[3], computed by removing the current non-dominated solutions from the solution set and computing the Pareto front of the remainder $k$ times.

In this work, we successfully used a simple filtering rule to remove non-converged solutions, relying on the fact that perfect fairness is difficult to achieve for non-trivial classifiers, and excluding all solutions with a fairness value greater than $1 - \epsilon$, with the value of $\epsilon$ set in the range of $10^{-3}$.

A common challenge in multi-objective optimization is an imbalance in the magnitude of different individual objective functions, as observed e.g. in Askin et al. (2025). In settings such as this, where

---

[3]Deb, K., Pratap, A., Agarwal, S., & Meyarivan, T. A. M. T. (2002). A fast and elitist multiobjective genetic algorithm: NSGA-II. IEEE transactions on evolutionary computation, 6(2), 182-197

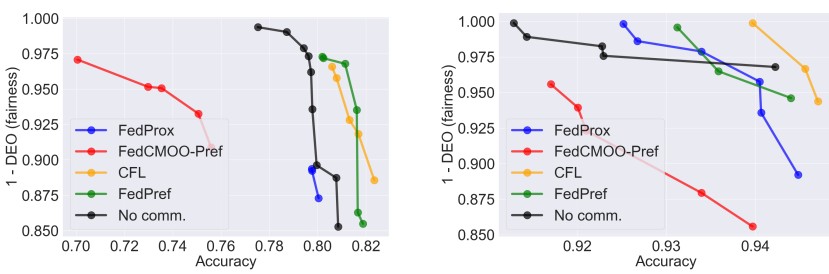

(a) Adult dataset with binarized race as the sensitive attribute and DEO fairness metric.

(b) Law School dataset with gender as the sensitive attribute and DEO metric.

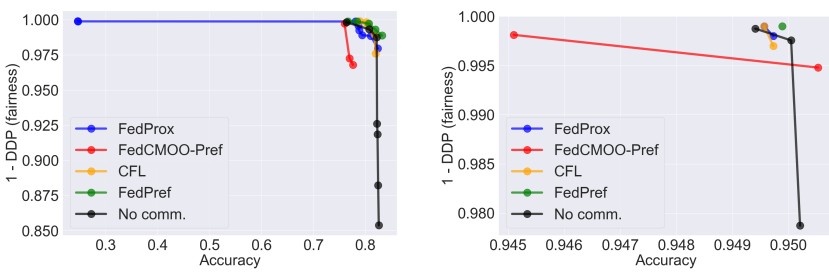

(c) Adult dataset with binarized race as the sensitive attribute and DDP fairness metric.

(d) Law School dataset with binarized race as the sensitive attribute and DDP metric.

Figure 5: Results of different algorithms on 10 clients on a selection of benchmark problems for accuracy and equality of opportunity (top row) and accuracy and demographic parity (bottom row). Clients were assigned heterogeneous preferences during each run, generated uniformly at random but the same across algorithms. Each point represents the output of a single client, with the Pareto fronts across all runs reported for each algorithm. All reported fairness metrics are inverted for ease of visualization, such that 1 corresponds to perfect fairness.

| Data | FedProx | CFL | FedCMOO | FedPref | no comm |
|---|---|---|---|---|---|
| Sens. Attr. | Acc,DEO | Acc,DEO | Acc,DEO | Acc,DEO | Acc,DEO |
| **Adult** | | | | | |
| Gender min | (80.0,80.1) | (80.7,80.8) | (73.8,77.8) | (80.4,80.8) | (79.0,79.5) |
| max | (83.7,87.6) | (84.7,85.0) | (84.2,87.0) | (85.6,86.1) | (81.7,85.3) |
| Race min | (79.8,80.0) | (80.6,80.8) | (70.5,75.7) | (80.8,80.8) | (77.5,79.6) |
| max | (87.8,91.9) | (85.9,88.9) | (94.4,96.5) | (89.8,89.8) | (86.8,90.6) |
| **Law School** | | | | | |
| Gender min | (91.8,93.2) | (92.6,92.7) | (91.5,93.4) | (91.7,93.3) | (91.5,91.7) |
| max | (83.4,94.8) | (86.8,88.6) | (83.1,98.6) | (84.2,89.6) | (84.8,86.8) |
| Race min | (92.8,93.5) | (92.8,93.0) | (91.9,92.4) | (92.8,93.2) | (91.6,91.7) |
| max | (70.2,75.2) | (69.3,79.8) | (65.6,74.6) | (70.3,79.9) | (71.6,72.7) |
| **Default** | | | | | |
| Gender min | (71.4,73.0) | (68.5,70.1) | (64.3,73.6) | (72.1,74.4) | (70.0,70.4) |
| max | (91.6,96.8) | (94.6,96.2) | (94.0,98.6) | (93.5,96.9) | (93.4,96.5) |

Table 8: Range of global performance results for accuracy and DEO on 10 clients with homogeneous preferences. Only results from the algorithm-specific Pareto front are reported (see also Fig. 2 in the main section). All values scaled by $10^2$. All reported fairness metrics are inverted for ease of visualization, such that 1 corresponds to perfect fairness.

| Data | FedProx | CFL | FedCMOO | FedPref | no comm |
|---|---|---|---|---|---|
| Sens. Attr. | Acc,DDP | Acc,DDP | Acc,DDP | Acc,DDP | Acc,DDP |
| **Adult** | | | | | |
| Gender min | (71.6,78.5) | (77.8,80.3) | (62.5,76.1) | (66.0,78.1) | (25.1,81.5) |
| max | (95.5,97.4) | (93.4,98.4) | (97.9,99.2) | (97.3,97.5) | (91.3,99.2) |
| Race min | (80.0,81.5) | (78.5,80.9) | (75.1,77.4) | (78.1,80.1) | (77.7,81.6) |
| max | (95.8,98.0) | (95.8,98.0) | (95.4,98.8) | (97.2,99.1) | (90.8,97.7) |
| **Law School** | | | | | |
| Gdr. min | (91.0,95.0) | (94.2,94.2) | (90.9,92.5) | (93.9,94.1) | (94.8,94.8) |
| max | (99.6,99.6) | (99.6, 99.6) | (99.2,99.5) | (99.6,99.6) | (99.5,99.5) |
| Race min | (93.1,93.1) | (94.4,94.8) | (94.5,94.9) | (93.8,94.6) | (94.4,94.8) |
| max | (97.9,97.9) | (99.0,99.6) | (99.2,99.6) | (97.9,98.1) | (98.9,99.1) |
| **Default** | | | | | |
| Gender min | (76.9,76.9) | (77.6,77.6) | (65.9,65.9) | (55.0,77.2) | (52.3,77.4) |
| max | (99.4,99.4) | (99.1,99.1) | (98.8,98.8) | (98.5,98.6) | (98.7,98.8) |

Table 9: Range of global performance results for accuracy and DDP (right) on 10 clients with homogeneous preferences. Only results from the algorithm-specific Pareto front are reported (see also Fig. 2 in the main section). All values scaled by $10^2$. All reported fairness metrics are inverted for ease of visualization, such that 1 corresponds to perfect fairness.

the potential values of the objective function are unbounded, an optimal mitigation strategy remains an open problem. Milojkovic et al. (2020) suggest normalizing objective functions by the initial values obtained for each. We note this approach tends to favor fairness over accuracy objectives, given that most fairness metrics produce near-perfect scores for the uniform predictions generated by untrained models. However, in practice this normalization appears to work quite well for fairness problems, both in Padh et al. (2021) and our own experiments. Other normalization approaches are possible.

Finally, converging with a high preference for fairness can be difficult, as the perfect fairness of an untrained model does not give a sufficient impetus for a model to start learning. Following the approach demonstrated by Padh et al. (2021), we mitigate this problem in our experiments by adding

|  | FedProx | CFL | FedCMOO | FedPref | no comm |
|---|---|---|---|---|---|
|  | Acc,DEO | Acc,DEO | Acc,DEO | Acc,DEO | Acc,DEO |
| **Adult** |  |  |  |  |  |
| Gender min | (80.0, 80.1) | (80.7, 80.8) | (73.8, 77.8) | (80.4, 80.8) | (79.0, 79.5) |
| max | (83.7, 87.6) | (84.7, 85.0) | (84.2, 87.0) | (85.6, 86.1) | (81.7, 85.3) |
| Race min | (79.8, 80.0) | (80.6, 80.8) | (70.5, 75.7) | (80.8, 80.8) | (77.5, 79.6) |
| max | (87.8, 91.9) | (85.9, 88.9) | (94.4, 96.5) | (89.8, 89.8) | (86.8, 90.6) |
| **Law School** |  |  |  |  |  |
| Gender min | (91.8, 93.2) | (92.6, 92.7) | (91.5, 93.4) | (91.7, 93.3) | (91.5, 91.7) |
| max | (83.4, 94.8) | (86.8, 88.6) | (83.1, 98.6) | (84.2, 89.6) | (84.8, 86.8) |
| Race min | (92.8, 93.5) | (92.8, 93.0) | (91.9, 92.4) | (92.8, 93.2) | (91.6, 91.7) |
| max | (70.2, 75.2) | (69.3, 79.8) | (65.6, 74.6) | (70.3, 79.9) | (71.6, 72.7) |
| **Default** |  |  |  |  |  |
| Gender min | (71.4, 73.0) | (68.5, 70.1) | (64.3, 73.6) | (72.1, 74.4) | (70.0, 70.4) |
| max | (91.6, 96.8) | (94.6, 96.2) | (94.0, 98.6) | (93.5, 96.9) | (93.4, 96.5) |

Table 10: Range of global performance results for accuracy and DEO with 10 clients on heterogeneous preferences. Only results from the algorithm-specific Pareto front are reported (see also Fig. 3 in the main section). All values scaled by $10^2$. All reported fairness metrics are inverted for ease of visualization, such that 1 corresponds to perfect fairness.

|  | FedProx | CFL | FedCMOO | FedPref | no comm |
|---|---|---|---|---|---|
|  | Acc,DDP | Acc,DDP | Acc,DDP | Acc,DDP | Acc,DDP |
| **Adult** |  |  |  |  |  |
| Gender min | (71.6, 78.5) | (77.8, 80.3) | (62.5, 76.1) | (66.0, 78.1) | (25.1, 81.5) |
| max | (95.5, 97.4) | (93.4, 98.4) | (97.9, 99.2) | (97.3, 97.5) | (91.3, 99.2) |
| Race min | (80.0, 81.5) | (78.5, 80.9) | (75.1, 77.4) | (78.1, 80.1) | (77.7, 81.6) |
| max | (95.8, 98.0) | (95.8, 98.0) | (95.4, 98.8) | (97.2, 99.1) | (90.8, 97.7) |
| **Law School** |  |  |  |  |  |
| Gender min | (91.0, 95.0) | (94.2, 94.2) | (90.9, 92.5) | (93.9, 94.1) | (94.8, 94.8) |
| max | (99.6, 99.6) | (99.6, 99.6) | (99.2, 99.5) | (99.6, 99.6) | (99.5, 99.5) |
| Race min | (93.1, 93.1) | (94.4, 94.8) | (94.5, 94.9) | (93.8, 94.6) | (94.4, 94.8) |
| max | (97.9, 97.9) | (99.0, 99.6) | (99.2, 99.6) | (97.9, 98.1) | (98.9, 99.1) |
| **Default** |  |  |  |  |  |
| Gender min | (76.9, 76.9) | (77.6, 77.6) | (65.9, 65.9) | (55.0, 77.2) | (52.3, 77.4) |
| max | (99.4, 99.4) | (99.1, 99.1) | (98.8, 98.8) | (98.5, 98.6) | (98.7, 98.8) |

Table 11: Range of global performance results for accuracy and DDP on 10 clients with heterogeneous preferences. Only results from the algorithm-specific Pareto front are reported (see also Fig. 3 in the main section). All values scaled by $10^2$. All reported fairness metrics are inverted for ease of visualization, such that 1 corresponds to perfect fairness.

a small fraction of the accuracy loss for regularization, e.g. for DDP loss:

$$\text{Loss}_{DDP} = \widehat{DDP} + 0.1 \cdot \text{BCELogitsLoss}. \qquad (6)$$

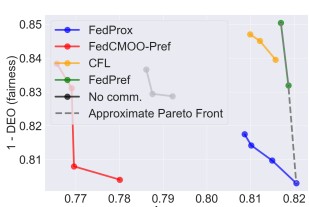

(a) Adult dataset with binarized gender as the sensitive attribute and DEO fairness metric.

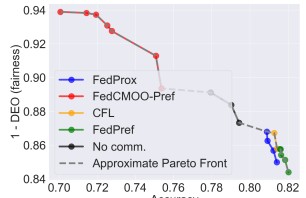

(b) Adult dataset with binarized race as the sensitive attribute and DEO fairness metric.

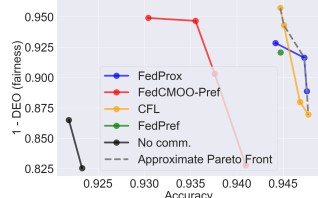

(c) Law School dataset with gender as the sensitive attribute and DEO metric.

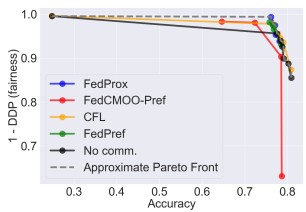

(d) Adult dataset with binarized gender as the sensitive attribute and DDP fairness metric.

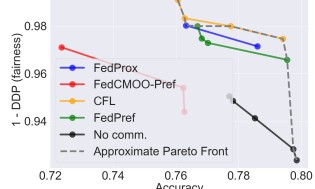

(e) Adult dataset with binarized race as the sensitive attribute and DDP fairness metric.

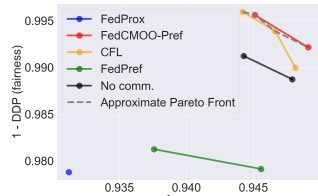

(f) Law School dataset with binarized race as the sensitive attribute and DDP metric.

Figure 6: Results of different algorithms on 50 clients on a selection of benchmark problems for accuracy and equality of opportunity (top row) and accuracy and demographic parity (bottom row). All clients were assigned the same preferences during a run, with 10 runs performed on preferences from $(0., 1.0)$ to $(0.9, 0.1)$, modified by steps of $(+0.1, -0.1)$. Each point represents the mean client output for a single run, with the Pareto fronts across all runs reported for each algorithm. All reported fairness metrics are inverted for ease of visualization, such that 1 corresponds to perfect fairness.

| Data - | Accuracy - DEO | | | | | Accuracy - DDP | | | | |
|---|---|---|---|---|---|---|---|---|---|---|
| Sens. attr. | FProx | CFL | FCMOO | FPref | no comm | FProx | CFL | FCMOO | FPref | no comm |
| Adult - G | 0.671 | 0.691 | 0.654 | **0.696** | 0.663 | 0.776 | **0.792** | 0.767 | 0.769 | 0.781 |
| Adult - R | 0.707 | 0.706 | 0.707 | 0.703 | **0.708** | 0.770 | **0.786** | 0.740 | 0.779 | 0.759 |
| Law - G | 0.880 | **0.907** | 0.893 | 0.870 | 0.799 | 0.939 | 0.940 | 0.926 | **0.941** | 0.906 |
| Law - R | 0.723 | 0.664 | 0.702 | 0.718 | **0.737** | **0.947** | 0.945 | 0.944 | 0.932 | 0.904 |
| Default - G | 0.759 | 0.733 | **0.799** | 0.746 | 0.684 | 0.667 | 0.669 | 0.627 | **0.684** | 0.672 |

Table 12: Hypervolumes of global performance results for accuracy and DEO (left) and accuracy and DDP (right) with 50 clients on homogeneous preferences. Higher is better (Fairness metrics are inverted, as in the results figures). Only results from the algorithm-specific Pareto front are reported (see also Fig. 6 and Fig. 7)

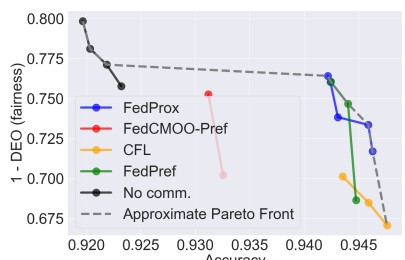

(a) Law school dataset with binarized race as the sensitive attribute and DEO fairness metric.

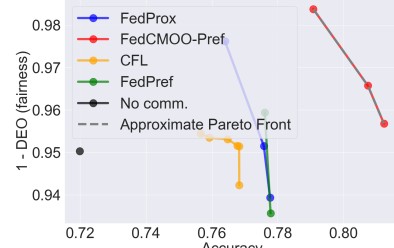

(b) Default dataset with gender as the sensitive attribute and DEO fairness metric.

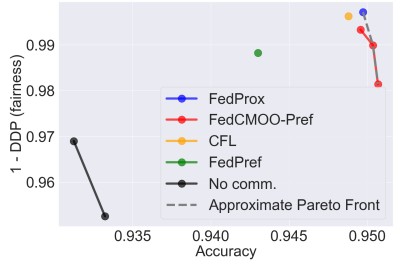

(c) Law School dataset with binarized race as the sensitive attribute and DDP fairness metric.

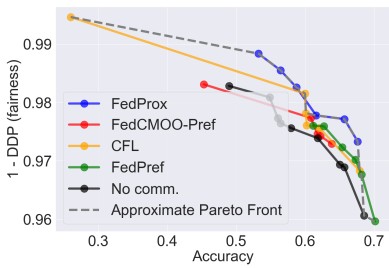

(d) Default dataset with gender as the sensitive attribute and DDP fairness metric.

Figure 7: Additional results of different algorithms on 50 clients on a selection of benchmark problems for accuracy and equality of opportunity (top row) and accuracy and demographic parity (bottom row). All clients were assigned the same preferences during a run, with 10 runs performed on preferences from $(0., 1.0)$ to $(0.9, 0.1)$, modified by steps of $(+0.1, -0.1)$. Each point represents the mean client output for a single run, with the Pareto fronts across all runs reported for each algorithm. All reported fairness metrics are inverted for ease of visualization, such that 1 corresponds to perfect fairness.

| | FedProx | CFL | FedCMOO | FedPref | no comm |
|---|---|---|---|---|---|
| | Acc,DEO | Acc,DEO | Acc,DEO | Acc,DEO | Acc,DEO |
| **Adult** | | | | | |
| Gender min | (80.9, 82.0) | (81.0, 81.6) | (76.6, 78.0) | (81.7, 81.9) | (78.6, 79.2) |
| max | (80.3, 81.7) | (84.0, 84.7) | (80.4, 83.8) | (83.2, 85.0) | (82.9, 83.7) |
| Race min | (80.9, 81.4) | (81.3, 81.4) | (70.1, 75.4) | (81.6, 82.0) | (77.9, 79.4) |
| max | (85.0, 86.8) | (85.8, 86.7) | (89.4, 93.9) | (84.4, 85.8) | (87.3, 89.1) |
| **Law School** | | | | | |
| Gender min | (94.4, 94.8) | (94.5, 94.8) | (93.0, 94.1) | (94.5, 94.5) | (92.2, 92.3) |
| max | (88.9, 92.8) | (87.0, 95.7) | (82.8, 94.9) | (92.1, 92.1) | (82.5, 86.5) |
| Race min | (94.2, 94.6) | (94.4, 94.8) | (93.1, 93.3) | (94.2, 94.5) | (92.0, 92.3) |
| max | (71.7, 76.4) | (67.1, 70.1) | (70.2, 75.3) | (68.6, 76.1) | (75.8, 79.8) |
| **Default** | | | | | |
| Gender min | (76.4, 77.8) | (75.7, 76.8) | (79.1, 81.2) | (77.6, 77.8) | (72.0, 72.0) |
| max | (93.9, 97.6) | (94.2, 95.4) | (95.7, 98.4) | (93.6, 95.9) | (95.0, 95.0) |

Table 13: Range of global performance results for accuracy and DEO on 50 clients with homogeneous preferences. Only results from the algorithm-specific Pareto front are reported (see also Fig. 6 and Fig. 7 in this appendix). All values scaled by $10^2$. All reported fairness metrics are inverted for ease of visualization, such that 1 corresponds to perfect fairness.

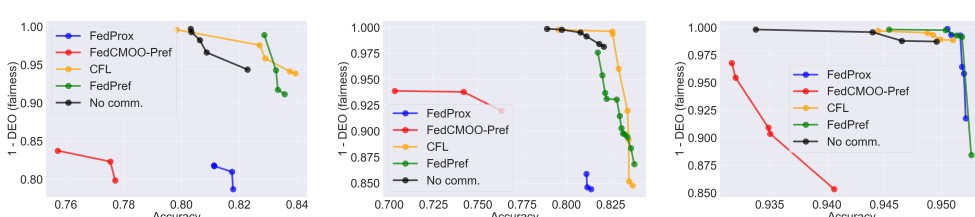

(a) Adult dataset with binarized gender as the sensitive attribute and DEO fairness metric.

(b) Adult dataset with binarized race as the sensitive attribute and DEO fairness metric.

(c) Law School dataset with gender as the sensitive attribute and DEO metric.

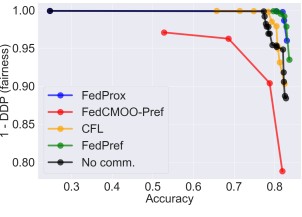
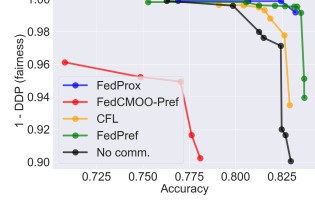
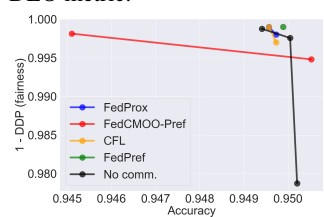

(d) Adult dataset with binarized gender as the sensitive attribute and DDP fairness metric.

(e) Adult dataset with binarized race as the sensitive attribute and DDP fairness metric.

(f) Law School dataset with binarized race as the sensitive attribute and DDP metric.

Figure 8: Results of different algorithms on 50 clients on a selection of benchmark problems for accuracy and equality of opportunity (top row) and accuracy and demographic parity (bottom row). Clients were assigned heterogeneous preferences during each run, generated uniformly at random but the same across algorithms. Each point represents the output of a single client, with the Pareto fronts across all runs reported for each algorithm. All reported fairness metrics are inverted for ease of visualization, such that 1 corresponds to perfect fairness.

|  | FedProx | CFL | FedCMOO | FedPref | no comm |
|---|---|---|---|---|---|
|  | Acc,DDP | Acc,DDP | Acc,DDP | Acc,DDP | Acc,DDP |
| **Adult** |  |  |  |  |  |
| Gender min | (76.1, 78.1) | (24.8, 80.8) | (64.6, 78.6) | (75.8, 78.4) | (24.9, 80.8) |
| max | (93.8, 99.4) | (87.3, 99.6) | (63.1, 98.3) | (93.1, 98.1) | (85.5, 99.5) |
| Race min | (76.3, 78.6) | (76.1, 79.4) | (72.4, 76.3) | (76.7, 79.5) | (77.7, 79.8) |
| max | (97.1, 98.0) | (97.5, 99.1) | (94.4, 97.1) | (96.6, 98.0) | (92.4, 95.1) |
| **Law School** |  |  |  |  |  |
| Gender min | (90.7, 94.2) | (94.5, 94.5) | (92.9, 92.9) | (94.7, 94.7) | (91.1, 92.1) |
| max | (98.7, 99.6) | (99.5, 99.5) | (99.7, 99.7) | (99.5, 99.5) | (97.8, 98.3) |
| Race min | (95.0, 95.0) | (94.9, 94.9) | (95.0, 95.1) | (94.3, 94.3) | (93.1, 93.3) |
| max | (99.7, 99.7) | (99.6, 99.6) | (98.1, 99.3) | (98.8, 98.8) | (95.3, 96.9) |
| **Default** |  |  |  |  |  |
| Gender min | (53.2, 67.6) | (26.0, 67.9) | (45.3, 63.9) | (61.2, 70.1) | (49.0, 68.6) |
| max | (97.3, 98.8) | (96.8, 99.5) | (97.3, 98.3) | (96.0, 97.6) | (96.1, 98.3) |

Table 14: Range of global performance results for accuracy and DDP on 50 clients with homogeneous preferences. Only results from the algorithm-specific Pareto front are reported (see also Fig. 6 and Fig. 7 in this appendix). All values scaled by $10^2$. All reported fairness metrics are inverted for ease of visualization, such that 1 corresponds to perfect fairness.

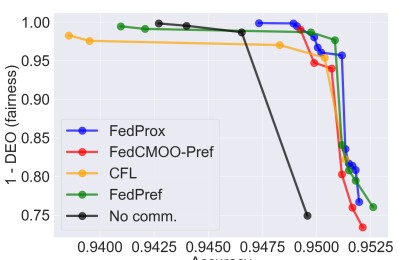

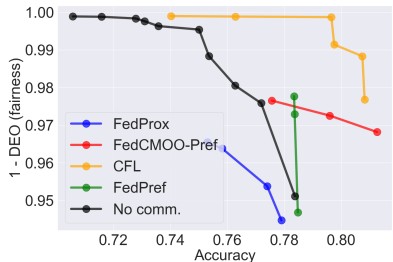

(a) Law school dataset with binarized race as the sensitive attribute and DEO fairness metric.

(b) Default dataset with gender as the sensitive attribute and DEO fairness metric.

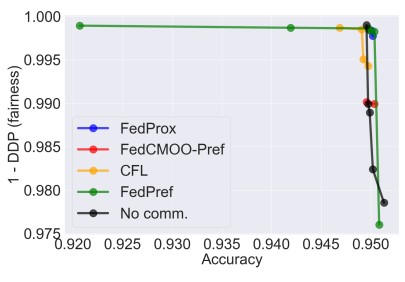

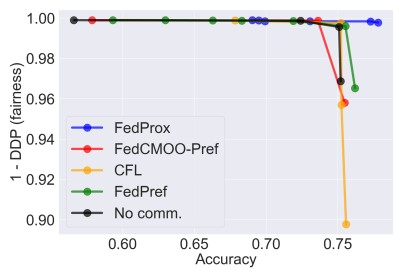

(c) Law School dataset with binarized race as the sensitive attribute and DDP fairness metric.

(d) Default dataset with gender as the sensitive attribute and DDP fairness metric.

Figure 9: Additional results of different algorithms on 50 clients on a selection of benchmark problems for accuracy and equality of opportunity (top row) and accuracy and demographic parity (bottom row). Clients were assigned heterogeneous preferences during each run, generated uniformly at random but the same across algorithms. Each point represents the output of a single client, with the Pareto fronts across all runs reported for each algorithm. All reported fairness metrics are inverted for ease of visualization, such that 1 corresponds to perfect fairness.

| Data - | Accuracy - DEO | | | | | Accuracy - DDP | | | | |
| Sens. attr. | FProx | CFL | FCMOO | FPref | no comm | FProx | CFL | FCMOO | FPref | no comm |
| Adult - G | 0.669 | **0.835** | 0.650 | 0.826 | 0.820 | 0.829 | 0.821 | 0.781 | **0.834** | 0.824 |
| Adult - R | 0.699 | **0.835** | 0.716 | 0.817 | 0.820 | 0.831 | 0.826 | 0.749 | **0.835** | 0.828 |
| Law - G | **0.951** | 0.948 | 0.910 | 0.951 | 0.948 | 0.941 | **0.950** | 0.921 | 0.950 | 0.949 |
| Law - R | **0.950** | 0.935 | 0.943 | 0.947 | 0.947 | 0.949 | 0.948 | 0.941 | 0.950 | **0.950** |
| Default - G | 0.752 | **0.807** | 0.793 | 0.767 | 0.782 | **0.777** | 0.754 | 0.753 | 0.761 | 0.751 |

Table 15: Hypervolumes of global performance results for accuracy and DEO (left) and accuracy and DDP (right) with 50 clients on heterogeneous preferences. Higher is better (Fairness metrics are inverted, as in the results figures). Only results from the algorithm-specific Pareto front are reported (see also Fig. 8 and Fig. 9)

| | FedProx | CFL | FedCMOO | FedPref | no comm |
|---|---|---|---|---|---|
| | Acc,DEO | Acc,DEO | Acc,DEO | Acc,DEO | Acc,DEO |
| **Adult** | | | | | |
| Gender min | (0.811, 0.787) | (0.798, 0.939) | (0.757, 0.799) | (0.829, 0.912) | (0.803, 0.944) |
| max | (0.818, 0.818) | (0.839, 0.996) | (0.777, 0.837) | (0.836, 0.989) | (0.823, 0.997) |
| Race min | (0.811, 0.844) | (0.795, 0.848) | (0.703, 0.920) | (0.818, 0.868) | (0.789, 0.982) |
| max | (0.814, 0.859) | (0.837, 0.998) | (0.763, 0.939) | (0.838, 0.976) | (0.821, 0.999) |
| **Law School** | | | | | |
| Gender min | (0.951, 0.918) | (0.944, 0.988) | (0.932, 0.853) | (0.945, 0.884) | (0.934, 0.987) |
| max | (0.952, 0.999) | (0.951, 0.997) | (0.941, 0.968) | (0.953, 0.998) | (0.950, 0.998) |
| Race min | (0.947, 0.767) | (0.939, 0.822) | (0.949, 0.734) | (0.941, 0.760) | (0.943, 0.749) |
| max | (0.952, 0.999) | (0.951, 0.983 | (0.952, 0.990) | (0.953, 0.994) | (0.950, 0.998) |
| **Default** | | | | | |
| Gender min | (0.753, 0.945) | (0.740, 0.977) | (0.776, 0.968) | (0.783, 0.947) | (0.706, 0.951) |
| max | (0.779, 0.965) | (0.808, 0.999) | (0.813, 0.977) | (0.785, 0.978) | (0.784, 0.999) |

Table 16: Range of global performance results for accuracy and DEO on $50$ clients with heterogeneous preferences. Only results from the algorithm-specific Pareto front are reported (see also Fig. 8 and Fig. 9 in the appendix). All values scaled by $10^2$. All reported fairness metrics are inverted for ease of visualization, such that 1 corresponds to perfect fairness.

| | FedProx | CFL | FedCMOO | FedPref | no comm |
|---|---|---|---|---|---|
| | Acc,DDP | Acc,DDP | Acc,DDP | Acc,DDP | Acc,DDP |
| **Adult** | | | | | |
| Gender min | (0.246, 0.960) | (0.657, 0.903) | (0.527, 0.788) | (0.797, 0.935) | (0.246, 0.884) |
| max | (0.830, 0.999) | (0.824, 0.999) | (0.819, 0.971) | (0.836, 0.999) | (0.828, 0.999) |
| Race min | (0.769, 0.992) | (0.791, 0.935) | (0.708, 0.902) | (0.753, 0.939) | (0.763, 0.900) |
| max | (0.832, 0.999) | (0.829, 0.997) | (0.781, 0.961) | (0.837, 0.998) | (0.830, 0.999) |
| **Law School** | | | | | |
| Gender min | (0.920, 0.993) | (0.894, 0.986) | (0.906, 0.997) | (0.898, 0.993) | (0.922, 0.989) |
| max | (0.943, 0.998) | (0.951, 0.999) | (0.922, 0.999) | (0.951, 0.999) | (0.950, 0.999) |
| Race min | (0.950, 0.998) | (0.947, 0.994) | (0.950, 0.990) | (0.921, 0.976) | (0.950, 0.979) |
| max | (0.950, 0.998) | (0.950, 0.999) | (0.950, 0.990) | (0.951, 0.999) | (0.951, 0.999) |
| **Default** | | | | | |
| Gender min | (0.690, 0.998) | (0.678, 0.898) | (0.579, 0.958) | (0.593, 0.965) | (0.566, 0.969) |
| max | (0.778, 0.999) | (0.756, 0.999) | (0.755, 0.999) | (0.762, 0.999) | (0.752, 0.999) |

Table 17: Range of global performance results for accuracy and DDP on $50$ clients with heterogeneous preferences. Only results from the algorithm-specific Pareto front are reported (see also Fig. 8 and Fig. 9 in this appendix). All values scaled by $10^2$. All reported fairness metrics are inverted for ease of visualization, such that 1 corresponds to perfect fairness.

