# OpenReview forum: "A New Class of Benchmarks for Federated Multi-Objective Learning"
_ICLR.cc/2026/Conference — Submitted to ICLR 2026_

### Official Review · Reviewer_8u8s · 2025-10-27

**Soundness:** 2
**Presentation:** 2
**Contribution:** 2
**Rating:** 2
**Confidence:** 4

**Summary:**

The paper argues that common multi-task learning (MTL) benchmarks (e.g., Multi-MNIST) are unsuitable for evaluating Federated Multi-Objective Learning (FMOL), since they lack genuine objective conflict under federation. The authors demonstrate this empirically, claiming that even simple FedAvg outperforms centralized multi-objective baselines. They then propose a new benchmark class based on fairness metrics (Demographic Parity, Equality of Opportunity, Equalized Odds), formulating differentiable approximations via tanh relaxations. Experiments with FedProx, CFL, FedCMOO, and FedPref across several fairness datasets aim to show that these fairness-based tasks better expose conflicting objectives in federated settings.

**Strengths:**

* highlights mismatch between centralized MOO/MTL and FMOL evaluation.
* Fairness metrics offer genuine conflicts; easy to port across models/objectives.
* Multiple FMOL baselines evaluated; implementation appears feasible to reproduce in principle.
* Both homogeneous and heterogeneous preference settings considered; includes 10- and 50-client studies.

**Weaknesses:**

Novelty/contribution:
* Incremental: fairness-as-objectives and fair-FL have prior art; no new algorithm or theory.
* No principled benchmark design framework (e.g., standardized partitioning, heterogeneity scales, or difficulty metrics); largely a collection of existing datasets plus a known relaxation.
* Central claim that MTL lacks inherent conflict under federation is not rigorously established (no gradient-conflict measures, task interference analysis, or counterexamples).

Soundness/experimental rigor:
* “FedAvg > centralized MOO” on Multi-MNIST not convincingly controlled: missing exact settings (model capacity, optimizer/epochs, learning rates), random seeds, CIs, and tuning parity; no ablations to isolate causes (architecture sharing, exploration from preference heterogeneity, etc.).
* Hypervolume reported without explicit reference point; fairness inversion (1 = fair) risks confusion; no statistical testing/variability reporting for HV or Pareto sets.
* Sensitivity to client partitioning and class/sensitive-attribute imbalance is acknowledged but not resolved; filtering of “near-perfect fairness” runs is ad hoc without robustness checks.
* Tanh-relaxation constant c and regularization mix (e.g., +0.1·BCE) lack principled selection; no stability/sensitivity results.

Presentation/clarity/grammar:
* Notation inconsistencies (e.g., y-hat placement), and a few typos (“is a is a well-known”); some equations/typesetting for fairness metrics are unclear and could confuse summation indices.
* Figures (especially the Multi-MNIST comparison) lack complete experimental context in captions; axes and normalization choices not fully explained; no error bars or CIs.
* Some references misdated (e.g., ProPublica “Machine Bias” typically 2016); fairness survey dates inconsistent; ensure consistent venue/year formatting.
* Appendices are long on grids/tables but short on interpretive diagnostics (e.g., why DDP harder than EO, when/why algorithms fail).

Scope/coverage:
* Evaluation limited to small tabular fairness datasets; no vision/NLP or larger-scale/federated-realism tasks.
* No alternative multi-objective families beyond fairness (e.g., calibration–accuracy, robustness–accuracy, latency–accuracy).

**Questions:**

1. How do you directly measure “conflict” in MTL vs FMOL (e.g., gradient cosine similarity across tasks, interference metrics, or conflict indices)?
1. What exact training budgets and seeds were used in Fig. 1; were centralized baselines tuned with the same search space/epochs/optimizers?
1. What is the hypervolume reference point per task/metric; how do results vary with that choice?
1. How is the tanh relaxation parameter c selected; show sensitivity and stability across datasets and objectives.
1. How robust are results to client partition strategies that preserve (label, sensitive) joint distributions; can you report replicated runs with CIs?
1. Can you add a principled difficulty measure (e.g., front curvature, dominated volume gap, gradient-conflict statistics) for the proposed benchmarks?

---

> ### Author Response · Authors · 2025-12-04
>
> We thank the reviewer for their comments. Our responses are presented below. Due to the length of the original review, comments have been split into two parts.
>
> We concur that significant work remains to be done; however, we believe that many of the aspects raised here are complex enough to require more work than can be fit into the scope of a single paper.
>
> *Incremental: fairness-as-objectives and fair-FL have prior art; no new algorithm or theory.*
>
> The proposed class of benchmarks is indeed based on existing widely available datasets and proven mathematical relations; we consider this to be a strength of the approach. The novelty of this contribution lies in (a) the identification of the weakness of established benchmarking approaches in federated multi-objective learning, and (b) the suggestion of readily available, scalable, and flexible benchmarks to remedy this weakness in existing and future frameworks. This will allow the rigorous evaluation of current and new FMOL algorithms on a standardized set of representative problems.
>
> *No principled benchmark design framework (e.g., standardized partitioning, heterogeneity scales, or difficulty metrics); largely a collection of existing datasets plus a known relaxation.*
>
> Prescribing standardized partitioning is not common practice in Federated Learning, as benchmarks are commonly adapted from the centralized setting and different FL problem settings require different partitioning choices. We agree that formal heterogeneity/difficulty metrics would be an interesting contribution to the field. We are not currently aware of the existence of applicable metrics to this setting, and believe the invention of such metrics to be out of the scope of this paper.
>
> *Central claim that MTL lacks inherent conflict under federation is not rigorously established (no gradient-conflict measures, task interference analysis, or counterexamples).*
>
> We agree that a wider range of experiments would strengthen this claim. However, we are not currently aware of the existence of well-established metrics to measure inherent task conflict, and we believe the invention of such metric to be out of the scope of this paper.
>
>
> **Soundness/experimental rigor:**
>
> *• “FedAvg > centralized MOO” on Multi-MNIST not convincingly controlled: missing exact settings (model capacity, optimizer/epochs, learning rates), random seeds, CIs, and tuning parity; no ablations to isolate causes (architecture sharing, exploration from preference heterogeneity, etc.). Hypervolume reported without explicit reference point; fairness inversion (1 = fair) risks confusion; no statistical testing/variability reporting for HV or Pareto sets.*
>
> We have added a description of hyperparameters for the motivating experiment and the reference point in the appendix. We agree that a deeper investigation into the causes behind the observed behaviour of FL on multi-task problems would be very interesting, but feel that this is out of the scope of this paper.
>
> *• Sensitivity to client partitioning and class/sensitive-attribute imbalance is acknowledged but not resolved; filtering of “near-perfect fairness” runs is ad hoc without robustness checks.*
>
> Indeed, we agree that this should be explored further. However, sensitivity to the partitioning of data is a fundamental aspect of federated learning paradigms, and different partitioning modes are frequently part of the experimental design for algorithmic validation.
>
> *• Tanh-relaxation constant c and regularization mix (e.g., +0.1·BCE) lack principled selection; no stability/sensitivity results.*
>
> As noted in the paper, this choice follows previous work in the field. Further exploring the implications of different variable choices would certainly be interesting, but arguably out of the scope of this work.

---

> > ### Author Response · Authors · 2025-12-04
> >
> > **Presentation/clarity/grammar:**
> > *• Notation inconsistencies (e.g., y-hat placement), and a few typos (“is a is a well-known”); some equations/typesetting for fairness metrics are unclear and could confuse summation indices.*
> >
> > We thank the reviewer for pointing out the existence of typos, and have corrected them. We did not find inconsistencies in y-hat placement; there may be confusion about the difference between the definitions given for y-hat (the predicted label) and y (the ground truth label). Beyond this, we are unsure what aspect of the formulation/typesetting of these equations is unclear.
> >
> > * • Figures (especially the Multi-MNIST comparison) lack complete experimental context in captions; axes and normalization choices not fully explained; no error bars or CIs.*
> >
> > Additional context has been added in the appendix. Error bars are not typically included in Pareto plots.
> >
> > *• Some references misdated (e.g., ProPublica “Machine Bias” typically 2016); fairness survey dates inconsistent; ensure consistent venue/year formatting.*
> >
> > We had cited a book where the original “Machine Bias” report had been published as a chapter. We have modified this to cite instead the original online report.
> >
> > *• Appendices are long on grids/tables but short on interpretive diagnostics (e.g., why DDP harder than EO, when/why algorithms fail).
> > Scope/coverage:
> >     • Evaluation limited to small tabular fairness datasets; no vision/NLP or larger-scale/federated-realism tasks.*
> >
> > We agree that larger/more complex benchmarking datasets would be interesting. However, this work is intended to supplement the typically used multi-task benchmarks that represent equally simplified tasks and are usually used as first-line evaluations of novel algorithms. As such, we feel that the construction and evaluation of complex benchmarks should be left to future work.
> >
> > *• No alternative multi-objective families beyond fairness (e.g., calibration–accuracy, robustness–accuracy, latency–accuracy).*
> >
> > We agree that such benchmarks would be interesting as well. However, we believe that their construction exceeds the scope of this work.
> >
> > **Questions:**
> >
> > Please see our in-line comments for reply to these questions.

---

### Official Review · Reviewer_43yz · 2025-10-31

**Soundness:** 3
**Presentation:** 3
**Contribution:** 2
**Rating:** 4
**Confidence:** 3

**Summary:**

This paper addresses the lack of suitable benchmarks for FMOL. The authors argue that existing multi-task benchmarks (e.g., Multi-MNIST, CelebA) often fail to exhibit genuine conflicts among objectives in federated contexts, as federated averaging can unintentionally eliminate objective trade-offs. To address this gap, the paper introduces a new class of benchmarks based on fairness-aware learning problems, where fairness metrics (e.g., demographic parity, equality of opportunity) serve as conflicting objectives against accuracy.

The authors propose a differentiable relaxation of fairness metrics (using tanh-based smoothing), allowing straightforward incorporation into stochastic optimization. They demonstrate this benchmark framework on several datasets (Adult, Law School, Credit Default), using multiple federated algorithms (FedProx, CFL, FedCMOO, FedPref) under both homogeneous and heterogeneous client preference settings. The experiments show that these fairness-based benchmarks yield nontrivial Pareto trade-offs and are more diagnostically rich than standard multi-task benchmarks.

**Strengths:**

The paper convincingly demonstrates that standard multi-task datasets (e.g., Multi-MNIST) may not produce meaningful trade-offs under federated aggregation, especially when clients have heterogeneous objectives. The motivating experiment is compelling and identifies an overlooked problem in current FMOL evaluation practices.
The paper provides nuanced observations — for example, that client preference heterogeneity might enhance parameter-space exploration — and discusses practical issues such as dataset imbalance and partitioning effects.
The authors evaluate four distinct FMOL algorithms under both homogeneous and heterogeneous preference settings, providing systematic comparisons. The analysis includes Pareto front visualizations and hypervolume metrics across multiple datasets and fairness metrics.

**Weaknesses:**

While the motivation is well-argued empirically, the paper lacks formal justification for why fairness-based conflicts are representative of general multi-objective tension in federated learning. A theoretical characterization of conflict strength or Pareto diversity across benchmark types would have strengthened the claim.

Only three datasets (Adult, Law School, Default) are used in the main experiments. Although Table 1 lists more possible datasets, it remains unclear how well the approach generalizes to non-tabular modalities (e.g., images, text), where fairness labels and sensitive attributes behave differently.

The paper does not report whether fairness-based trade-offs are preserved or amplified by federation compared to centralized multi-objective training. This comparison would help confirm that the federated context indeed adds complexity.

Since fairness datasets involve sensitive demographic data, the paper could have addressed privacy-preserving concerns or how benchmark construction avoids ethical pitfalls. This is relevant for federated settings.

**Questions:**

why certain algorithms (e.g., FedPref vs. CFL) succeed or fail on the new benchmarks


Since different fairness metrics are often incompatible, how sensitive are the benchmark results to the choice of metric (DDP vs. DEO)?

---

> ### Author Response · Authors · 2025-12-04
>
> We thank the reviewer for their comments. We reply to the weaknesses raised by this review in detail below.
>
> *While the motivation is well-argued empirically, the paper lacks formal justification for why fairness-based conflicts are representative of general multi-objective tension in federated learning. A theoretical characterization of conflict strength or Pareto diversity across benchmark types would have strengthened the claim.*
>
> Previous results from the field of fair machine learning show the existence of conflicts, as discussed e.g. in Castelnovo et al., “A clarification of the nuances in the fairness metrics landscape”.
>
> *Only three datasets (Adult, Law School, Default) are used in the main experiments. Although Table 1 lists more possible datasets, it remains unclear how well the approach generalizes to non-tabular modalities (e.g., images, text), where fairness labels and sensitive attributes behave differently.*
>
> The existence of conflicts in principle is independent of the format of data. This class of benchmarks indeed extends across different types of datasets representing different problem modalities, which we regard as an advantage of this approach.
>
> *The paper does not report whether fairness-based trade-offs are preserved or amplified by federation compared to centralized multi-objective training. This comparison would help confirm that the federated context indeed adds complexity.*
>
> This work is concerned with the complexity introduced into the federated learning scheme by multi-objective heterogeneity. The difficulty of the underlying MO problem itself should not be affected by the use of federated learning. This work is based on the theoretical knowledge of existing conflicts, and the experiments in the paper demonstrate that conflict in practice as well. This is in contrast to the motivating example, where we suspect that little to no underlying conflict exists.
>
> *Since fairness datasets involve sensitive demographic data, the paper could have addressed privacy-preserving concerns or how benchmark construction avoids ethical pitfalls. This is relevant for federated settings.*
>
> Privacy is indeed a relevant challenge and one that FL was designed for. However, these proposed benchmarks rely on existing datasets in the public domain, and so do not risk exposing sensitive information. Risks arising from the application of fairness methods to novel datasets should certainly be considered, but are out of the scope of this work establishing benchmarks on public datasets.

---

### Official Review · Reviewer_4buB · 2025-11-01

**Soundness:** 2
**Presentation:** 3
**Contribution:** 2
**Rating:** 4
**Confidence:** 3

**Summary:**

This paper addresses the underexplored problem of benchmarking Federated Multi-Objective Learning. The authors argue that widely used multi-task benchmarks such as Multi-MNIST fail to exhibit genuine objective conflicts once deployed in a federated setting, leading to artificially easy problems. To remedy this, they propose a new class of fairness-based benchmarks, derived from established group fairness metrics. These benchmarks combine fairness and accuracy objectives, creating natural trade-offs. The authors evaluate several algorithms under homogeneous and heterogeneous preference settings.

**Strengths:**

The paper makes an interesting observation: existing multi-task benchmarks collapse under federated training, masking genuine multi-objective conflicts. This is an insightful finding that challenges standard evaluation practice. The proposed fairness-based benchmarks are well motivated by the inherent trade-off between fairness and accuracy. The experiments are competently executed across several algorithms and preference settings, offering initial evidence that the new benchmarks better reflect real multi-objective challenges. The discussion of homogeneous vs. heterogeneous preferences is also interesting.

**Weaknesses:**

While the paper presents a very interesting observation in Section 3 that federated training on Multi-MNIST under heterogeneous preferences may remove apparent task conflicts, but the analysis remains speculative. The experiment is limited because the Multi-MNIST setup is too simplistic to convincingly support the broader claim. More realistic examples or, ideally, a theoretical analysis explaining why preference heterogeneity mitigates conflicts in federation would greatly strengthen the argument.

Moreover, as a benchmark paper, some datasets used (Adult, Law School, Credit Default) are rather dated and small in scale. It is therefore unclear whether the proposed benchmarks generalize to realistic, large-scale FMOL settings.

Finally, the experimental comparison is quite narrow: only four algorithms are evaluated, and none of the more recent state-of-the-art multi-objective optimization methods are included. This limited scope weakens the empirical validation of the proposed benchmarks’ difficulty and generality.

**Questions:**

Please refer to weaknesses

---

> ### Author Response · Authors · 2025-12-04
>
> We are grateful to the reviewer for their considered and insightful comments. We respond to the weaknesses raised by this review in the comments below.
>
> *While the paper presents a very interesting observation in Section 3 that federated training on Multi-MNIST under heterogeneous preferences may remove apparent task conflicts, but the analysis remains speculative. The experiment is limited because the Multi-MNIST setup is too simplistic to convincingly support the broader claim. More realistic examples or, ideally, a theoretical analysis explaining why preference heterogeneity mitigates conflicts in federation would greatly strengthen the argument.*
>
> The need for extending the motivating experiments is raised in several reviews, and we will aim to add more. We feel that a full experimental and theoretical exploration of the mechanism for this observed behaviour would exceed the scope of this paper and should be addressed in a separate work. (As an initial intuition, we suspect that the behaviour is related to the greater exploration of the parameter space forced by locally diverging models – some existing multi-task solving algorithms appear to rely on a similar mechanism by adjusting preferences during the learning process.)
> Given the current prevalence of and singular focus on multi-task problems to benchmark multi-objective problems, we believe there is merit in highlighting this issue to the wider community expediently, and offering an initial alternative.
>
> *Moreover, as a benchmark paper, some datasets used (Adult, Law School, Credit Default) are rather dated and small in scale. It is therefore unclear whether the proposed benchmarks generalize to realistic, large-scale FMOL settings.*
>
> We agree that exploring the extension of benchmarks to larger-scale datasets and problems is very important; however, this initial work was developed with the intention of expediently supplementing/replacing other small-scale, simplified benchmarking datasets such as the Multi-MNIST benchmark discussed in the motivating example. As such, we believe that the small size and wide availability of this dataset has merit for the intended use case, and we would prefer relegating the discussion of realistic large-scale benchmarks to other work.
>
> *Finally, the experimental comparison is quite narrow: only four algorithms are evaluated, and none of the more recent state-of-the-art multi-objective optimization methods are included. This limited scope weakens the empirical validation of the proposed benchmarks’ difficulty and generality.*
>
> We are not aware of other recent federated multi-objective algorithms apart from FedMGDA, FedCMOO and FedPref. As FedCMOO is a direct extension of FedMGDA, this was dropped. (Other combinations of FL and MO methods in the literature largely use MO methods to improve the FL algorithms, unrelated to the problem that is solved by the clients.)
> To provide a wider base of comparison, our experiment design also contained additional standard heterogeneity-mitigating algorithms that are not designed explicitly for the multi-objective setting.

---

### Meta-Review · Area_Chair_Rkmf · 2026-01-03

**Summary:**

The three reviewers’ comments are largely consistent and raise concerns regarding:
(1) limitations of the problem formulation and experimental setup, as well as the lack of formal justification;
(2) the use of outdated and small-scale datasets for a benchmark paper;
(3) the limited scope of experimental evaluation in terms of the FMOL algorithms tested; and
(4) the absence of a principled framework for benchmark design.

In addition, the reviewers raise several questions concerning technical details and presentation clarity.

AC comments. To assess the authors’ claim that two out of three reviews were fully generated by large language models, the AC carefully examined the abstract and the first three sections of the paper. As it stands, the paper appears ill-motivated for several reasons:

1. Ambiguity of “benchmark”. In a generic sense, the term “benchmark” could refer to performance benchmarks, algorithm benchmarks, model benchmarks, or dataset benchmarks. However, even after reading the abstract and the first two sections, it remains unclear what specific notion of “benchmark” the authors intend to address.

2. Questionable benchmark construction. In this paper, “benchmark” refers to constructing a new class of multi-objective (multi-criteria) problems—potentially with inherent objective conflicts—and then using these problems to evaluate FMOL algorithms. This approach is difficult to justify from a scientific standpoint.

3. Reversal of the standard scientific workflow. Typically, problems arise from nature or society and are then abstracted and formalized to make them tractable; algorithms are subsequently designed to solve these problems. By contrast, the approach adopted in this paper—starting from existing algorithms and constructing problems tailored to them—is uncommon and lacks a clear methodological justification.

**Reviewer Concerns:**

Except several questions concerning technical details and presentation clarity, major concerns above are still outstanding.

**Reviewer Scores:**

No score would likely be changed.

---

### Decision · Program_Chairs · 2026-01-26

Reject